# ZODIAC: A CARDIOLOGIST-LEVEL LLM FRAMEWORK FOR MULTI-AGENT DIAGNOSTICS

## ABSTRACT

Large language models (LLMs) have demonstrated remarkable progress in health-care. However, a significant gap remains regarding LLMs' professionalism in domain-specific clinical practices, limiting their application in real-world diagnostics. In this work, we introduce ZODIAC, an LLM-powered framework with cardiologist-level professionalism designed to engage LLMs in cardiological diagnostics. ZODIAC assists cardiologists by extracting clinically relevant characteristics from patient data, detecting significant arrhythmias, and generating preliminary reports for the review and refinement by cardiologists. To achieve cardiologist-level professionalism, ZODIAC is built on a multi-agent collaboration framework, enabling the processing of patient data across multiple modalities. Each LLM agent is fine-tuned using real-world patient data adjudicated by cardiologists, reinforcing the model's professionalism. ZODIAC undergoes rigorous clinical validation with independent cardiologists, evaluated across eight metrics that measure clinical effectiveness and address security concerns. Results show that ZODIAC outperforms industry-leading models, including OpenAI's GPT-4o, Meta's Llama-3.1-405B, and Google's Gemini-pro, as well as medical-specialist LLMs like Microsoft's BioGPT. ZODIAC demonstrates the transformative potential of specialized LLMs in healthcare by delivering domain-specific solutions that meet the stringent demands of medical practice. Notably, ZODIAC has been successfully integrated into electrocardiography (ECG) devices, exemplifying the growing trend of embedding LLMs into Software-as-Medical-Device (SaMD).

## 1 INTRODUCTION

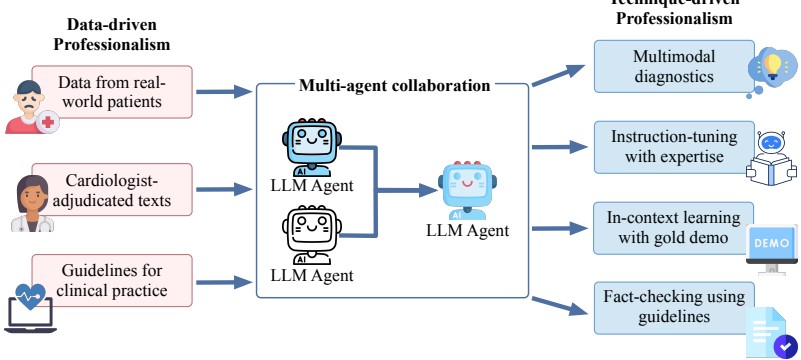

Figure 1: ZODIAC attains cardiologist-level professionalism through a combination of advanced data integration with sophisticated technical designs.

As technology continues to revolutionize healthcare, artificial intelligence (AI) has emerged as a crucial component of medical devices, driving the expansion of *digital health* in clinical practice (FDA, 2020). Among the most promising AI advancements, large language models (LLMs) are unlocking new possibilities in digital health. With their human-like conversational skills and vast pre-trained knowledge, LLMs are increasingly being adopted as clinical support tools by industry

leaders, evolving into specialized clinical agents (Boonstra et al., 2024; Gala & Makaryus, 2023; Xu et al., 2024). This evolution has given rise to industrial products such as Microsoft's BioGPT (Luo et al., 2022), Google's Med-Gemini (Saab et al., 2024) and Med-PaLM (Tu et al., 2024), as well as a range of open-source medical specialist LLMs (Chen et al., 2023a; 2024b; ContactDoctor, 2024; Wang et al., 2024c) built on Meta's Llama (Touvron et al., 2023).

Despite these advancements, integrating LLMs into real-world healthcare practice remains in its early stages, where a significant gap exists in their **professionalism** (Davenport & Kalakota, 2019; Asan et al., 2020; Weber et al., 2024; Quinn et al., 2022). Bridging these gaps is critical, especially when deploying LLMs in healthcare settings governed by the FDA's Software-as-a-Medical-Device (SaMD) regulations (FDA, 2018). These regulations require software to demonstrate expert-level proficiency to function as a clinical assistant. However, current LLMs often fall short of this standard due to their general-purpose design, which lacks alignment with the specific standards of clinical practice (Khan et al., 2023; Wang et al., 2021; Kerasidou et al., 2022). In the context of SaMD, LLMs need not have universal capabilities but they must perform specialized tasks with the professionalism and accuracy expected in life-critical healthcare environments (Kelly et al., 2019; Yan et al., 2023). Achieving this alignment is vital to ensure LLMs meet the stringent requirements of real-world healthcare deployment.

**Our Work.** This study aims to address the challenge of aligning LLMs with the SaMD practice in the field of *Cardiology*, focusing on the clinical findings and interpretation of electrocardiogram (ECGs) (Yanowitz, 2012). We introduce ZODIAC, an LLM-powered, multi-agent framework designed to achieve cardiologist-level professionalism. ZODIAC assists cardiologists by identifying clinically relevant characteristics from patient data, detecting significant arrhythmias, and generating preliminary reports for expert review and refinement (details in 3). As illustrated in Figure 1, ZODIAC combines advanced data integration with sophisticated technical designs to achieve professional performance. Specifically:

**I) Data-Driven Professionalism:** ZODIAC is built on **real-world data**, including (1) patient data collected from clinics, (2) cardiologist-adjudicated texts, and (3) clinical guidelines. This ensures professionalism in two key aspects: First, ZODIAC captures real-world cardiological characteristics, such as arrhythmias and their contributing factors, rather than relying on benchmarks or synthetic datasets that may not reflect clinical realities. Second, direct involvement from human experts (cardiologists) ensures ZODIAC is fine-tuned to match expert-level performance, while adherence to clinical guidelines mitigates potential biases or errors, enhancing diagnostic accuracy and safety.

**II) Technique-Driven Professionalism:** The technical design of ZODIAC aligns with cardiologist-level diagnostic practices. Our pipeline is **multi-agent**, utilizing multiple LLMs to analyze **multi-modal** patient data, including clinical metrics in tabular format and ECG tracings in image format. This multi-agent framework represents a paradigm cardiologists use to identify key characteristics and interpret findings for clinically significant arrhythmias. Furthermore, during fine-tuning and inference, we employ cardiologist-adjudicated data, integrating multimodal diagnostic professionalism through **instruction tuning** and **in-context learning**. Instruction tuning embeds professionalism into the LLM's parameters, while in-context learning provides professional demonstrations to further reinforce ZODIAC's diagnostics. Finally, we incorporate **fact-checking** against established cardiological guidelines to ensure the system generates accurate, expert-verified diagnostics.

**Clinical Validation.** Our experiments aim to bridge the gap between cardiologists and LLM-powered systems in terms of recognizing the professionalism of LLMs. To this end, we conduct a series of clinical validations to assess ZODIAC's clinical effectiveness and security. We utilize eight evaluation metrics and collaborate with cardiologists to evaluate ZODIAC's performance against leading LLMs, including OpenAI's ChatGPT-4o, Google's Gemini-Pro, Meta's Llama-405B, and specialized medical LLMs like Microsoft's BioGPT and Llama variants. With cardiologists endorsing ZODIAC through its application in real patient diagnostics, we underscore its practical utility from the healthcare provider's perspective.

**Blueprint and Lifecycle Prospect.** ZODIAC has been successfully deployed on Amazon AWS and integrated with clinical settings to assist in cardiological diagnostics (details in 4.4). The design, development, and deployment of ZODIAC provide a comprehensive blueprint for introducing professional-grade LLM agents into clinical-grade SaMD, encompassing data utilization, expert-level technical pipelines, and clinical validation. Furthermore, we incorporate human expertise

(cardiologists) throughout data preparation and clinical validation, ensuring the professionalism of the proposed system while earning expert endorsement and trust through real-world validation. This approach promotes establishing a "humans-in-the-loop" lifecycle in responsible AI development (Food et al., 2021).

In summary, this work makes the following contributions:

- We introduce ZODIAC, a cardiologist-level, multi-agent framework for patient-specific diagnostics, representing a significant step forward in aligning LLM capabilities with professional software-as-medical-device (SaMD) standards.

- We provide a comprehensive blueprint for constructing ZODIAC, offering a scalable framework that can guide the development of clinical-grade LLM agents across various clinical domains.

- Through rigorous clinical validation, we demonstrate the effectiveness of ZODIAC while establishing a model for integrating human oversight throughout the AI lifecycle, crucial for promoting responsible AI development under human regulation.

## 2 RELATED WORK

**LLMs in Clinical Diagnostics.** LLMs have shown considerable progress in processing and interpreting vast amounts of unstructured medical data, such as patient records, medical literature, and diagnostic reports. For example, Han et al. (2024) introduced a system that automatically summarizes clinical notes during interactions between patients and clinicians, while Ahsan et al. (2023) explored the role of LLMs in retrieving key evidence from electronic health records (EHRs). Despite these successes, concerns persist regarding LLMs' domain-specific expertise and professional performance in high-stakes, life-critical clinical settings (Nashwan & AbuJaber, 2023; Jahan et al., 2024; Wang et al., 2024a; Li et al., 2024). This work addresses these concerns by designing and validating ZODIAC through our design and experiments specifically for cardiological diagnostics.

**Multi-Agent Frameworks.** Multi-agent frameworks have been extensively studied to enhance LLM capabilities in handling complex tasks and managing distributed processes (Wang et al., 2024b; Hong et al., 2023; Du et al., 2023; Chan et al., 2023). In healthcare, where collaboration across different expertise is essential, multi-agent frameworks have shown their potential in optimizing patient management, coordinating care between various agents (e.g., doctors, nurses, administrative systems), and supporting decision-making processes (Furmankiewicz et al., 2014; Jemal et al., 2014; Shakshuki & Reid, 2015). Recent studies have also focused on leveraging multi-LLM agents to reduce manual tasks in healthcare workflows. For instance, Chen et al. (2024a) employed ChatGPT in distinct roles within a coordinated workflow, to automate tasks like database mining and drug repurposing, while ensuring quality control through role-based collaboration.

**Cardiological Diagnostic Systems.** Current cardiological diagnostic systems primarily depend on rule-based algorithms or single-agent approaches for identifying cardiovascular risk factors or predicting cardiac events (Goff Jr et al., 2014; Sud et al., 2022; Olesen et al., 2012). In recent years, deep learning models have been introduced into cardiology (Hannun et al., 2019; Acharya et al., 2019). However, there remains a significant gap in incorporating recent LLMs into cardiological diagnostics—a gap that this work addresses significantly.

## 3 PROBLEM FORMULATION FROM CARDIOLOGIST-LEVEL DIAGNOSTICS

This section outlines how ZODIAC is aligned with cardiological diagnostics. Section 3.1 defines the task of cardiological diagnostics and its key components. Section 3.2 introduces the multimodal data used in real-world diagnostics. Finally, in Section 3.3, we formalize the task from the perspective of LLMs using a multi-agent framework.

### 3.1 THE DIAGNOSTIC TASK AND ITS KEY COMPONENTS

The focus of this paper is the detection of clinically significant arrhythmias using patient data. We categorize the key components into two main categories: patient data and diagnostic outputs.

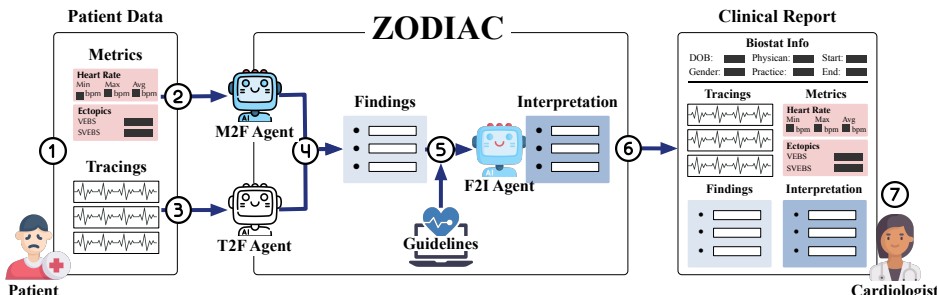

Figure 2: ZODIAC aligns with cardiological practice through a multi-agent framework that integrates patient data across various modalities: ① Patient data is collected in two modalities: tabular metrics and ECG tracings (images). ② A metrics-to-findings LLM agent processes the tabular metrics and generates text-based clinical findings. ③ An tracings-to-findings LLM agent analyzes the ECG tracings to produce additional text-based clinical findings. ④ The clinical findings from both agents are then combined. ⑤ A findings-to-interpretation LLM agent synthesizes these findings with clinical guidelines into comprehensive diagnostic interpretation. ⑥ ZODIAC generates a patient-specific report by integrating the metrics, tracings, clinical findings, and diagnostic interpretation. ⑦ A cardiologist validates the quality of the generated findings and interpretations (details in 5). For simplicity, we omit the biostatistics ($\mathcal{B}$) in this figure, which is considered in steps ①②③ by default.

**Patient Data** is comprised of three sections: (1) Biostatistical information ($\mathcal{B}$) provides background details about the patient such as date of birth, gender, and age group. (2) Metrics ($\mathcal{M}$) summarize cardiological attributes and their corresponding values presented in a tabular format, providing an overview of 24-hour monitored statistics for the patient. For example, *AF Burden: 12%* indicates that the patient experienced atrial fibrillation for 12% of the whole monitoring period. (3) Tracings ($\mathcal{T}$) includes ECG images depicting clinically significant arrhythmias such as AFib/Flutter (Atrial Fibrillation / Atrial Flutter), Pause, VT (Ventricular Tachycardia), SVT (Supraventricular Tachycardia), and AV Block (Atrioventricular Block). $\mathcal{T}$ presents a concise but representative segment of the 24-hour monitoring, such as a 10-second strip highlighting the highest degree of AV block.

**Diagnostic Outputs** is comprised of two elements: clinical Findings ($\mathcal{F}$) and Interpretation ($\mathcal{I}$), both presented as expert-crafted natural language statements by cardiologists. $\mathcal{F}$ outlines key observations directly from clinically relevant characteristics, while $\mathcal{I}$ provides the final diagnostics, interpreting these findings. For example, the finding *PR Interval is 210 milliseconds in the ECG tracings* leads to the interpretation: *The PR interval is slightly prolonged, suggesting a first-degree AV block.*

Once $\mathcal{F}$ and $\mathcal{I}$ are completed by cardiologists (or by ZODIAC), a clinical end-of-study report is generated for the patient, including $(\mathcal{B}, \mathcal{M}, \mathcal{T}, \mathcal{F}, \mathcal{I})$, as illustrated in the right part of Figure 2.

## 3.2 CARDIOLOGICAL DIAGNOSTICS WITH MULTIMODAL DATA

Cardiological diagnostics follows a process: $(\mathcal{B}, \mathcal{M}, \mathcal{T}) \rightarrow \mathcal{F} \rightarrow \mathcal{I}$. In addition, the final interpretation is guided by clinical guidelines, denoted as $\mathcal{G}$, which are consensus-based recommendations designed to support healthcare providers in making evidence-based decisions (detailed in D).

A cardiologist begins by reviewing the patient's data $(\mathcal{B}, \mathcal{M}, \mathcal{T})$ to identify clinically relevant characteristics, such as the *PR interval*, which are key for diagnosing arrhythmias. These identified characteristics are then summarized into natural language statements, referred to as findings $\mathcal{F}$, which integrate insights from both tabular metrics $\mathcal{M}$ and image-based ECG tracings $\mathcal{T}$. For example, the *PR interval* is derived from $\mathcal{T}$, while the *AF burden* is obtained from $\mathcal{M}$. Finally, cardiologists synthesize the findings $\mathcal{F}$ with their clinical expertise and the established guidelines $\mathcal{G}$ to form the final interpretation $\mathcal{I}$.

## 3.3 PROBLEM FORMULATION IN ZODIAC

As illustrated in Figure 2, ZODIAC represents the diagnostic process through a multi-agent collaboration. Each LLM agent is tasked with a specific stage in the diagnostic workflow, enhancing the

system's ability to identify hybrid characteristics across multiple modalities. ZODIAC is composed of three agents:

1. Metrics-to-Findings Agent ($\theta_{\text{M2F}}$): A table-to-text LLM that extracts key characteristics from tabular metrics ($\mathcal{M}$), while incorporating patient biostatistics from $\mathcal{B}$ to generate clinical findings.

2. Tracings-to-Findings Agent ($\theta_{\text{T2F}}$): An image-to-text LLM that identifies key factors from ECG tracings ($\mathcal{T}$), integrates relevant information from $\mathcal{B}$, and produces clinical findings.

3. Findings-to-Interpretation Agent ($\theta_{\text{F2I}}$): A text-based LLM that synthesizes findings ($\mathcal{F}$) from both the $\theta_{\text{M2F}}$ and $\theta_{\text{T2F}}$, applies clinical guidelines from $\mathcal{G}$, and generates the interpretation ($\mathcal{I}$).

Formally, the process of ZODIAC is formulated as:

$$\mathcal{I} \leftarrow \theta_{\text{F2I}}(\mathcal{F}, \mathcal{G}) \quad s.t. \quad \mathcal{F} \leftarrow \theta_{\text{M2F}}(\mathcal{M}, \mathcal{B}) \cup \theta_{\text{T2F}}(\mathcal{T}, \mathcal{B}) \tag{1}$$

In this process, $\theta_{\text{M2F}}$ and $\theta_{\text{T2F}}$ independently generate clinical findings based on $\mathcal{M}$ and $\mathcal{T}$, respectively, which are then combined to form $\mathcal{F}$. This approach adheres to cardiological diagnostics as each finding in $\mathcal{F}$ corresponds to evidence derived from a specific modality – either metrics or ECG tracings.

## 4 DESIGN, DEVELOPMENT, AND DEPLOYMENT OF ZODIAC

This section details how ZODIAC achieves cardiologist-level expertise, as well as its compliance with Software-as-a-Medical-Device (SaMD) regulations. We begin by discussing the process of real-world data collection and the professionalism-incorporated curation, outlined in Section 4.1. Next, we introduce the instruction fine-tuning in Section 4.2, where the curated data is used to imbue the LLM agents with domain-specific expertise. During inference, as described in Section 4.3, we leverage in-context learning and fact-checking to enhance diagnostic professionalism through collaborative multi-agent interactions. Finally, we present our approach to SaMD-compliant deployment in Section 4.4.

### 4.1 DATA COLLECTION AND PROFESSIONALISM-INCORPORATED CURATION

Our data is characterized as *real*, *representative*, and *professionalism-incorporated*.

**Real-World Patient Data.** Instead of relying on public benchmarks from third-party or synthetic data—which often raise concerns about trustworthiness or misalignment with specific clinical applications (Chouffani El Fassi et al., 2024; Fehr et al., 2024; Youssef et al., 2024)—we utilized ECG data sourced from our collaborating healthcare institutions[1] **under an IRB-approved protocol**, with removed patient identifiers to ensure privacy protection. The raw data collection consists of tabular metrics ($\mathcal{M}$) and ECG tracings ($\mathcal{T}$), as depicted in the left part of Figure 2. To ensure the clinical relevance, we engaged five independent cardiologists to review the data, resulting in a final dataset of 2,000+ patients. Of these, 5% were used for clinical validation (Section 5), while the remainder were used for fine-tuning (Section 4.2).

**Representative Groups.** Following FDA's guideline (Food et al., 2021), it is crucial to include representative data, not simply to amass large volumes. Our dataset encompasses comprehensive arrhythmia types and ensures balanced representation across age and gender demographics, as detailed in Figure 3-(d).

**Incorporating Cardiologist-Level Professionalism.** When reviewing the raw data, cardiologists are asked to write professional findings ($\mathcal{F}$) and interpretation ($\mathcal{I}$) in accordance with established clinical guidelines ($\mathcal{G}$). This process facilitates fine-tuning the LLMs, embedding cardiologist-level reasoning, evidence-based statements, and a structured format into the models. To optimize the cardiologists' time, medical research assistants first draft the $\mathcal{F}$ and $\mathcal{I}$, which are subsequently reviewed and independently adjudicated by the cardiologists. Additionally, each cardiologist randomly audits at least 50% of their peers' drafts to address issues such as incompleteness, inconsistency, or diagnostic inaccuracies. This peer-review process not only improves the data quality but also ensures standardized findings and interpretation with professional accuracy.

---

[1] For anonymity, the names of our collaborating institutions are withheld but will be disclosed upon publication of this paper.

Figure 3: (a)-(c) illustrate the prompts used for $\theta_{\texttt{M2F}}$ (prompts for $\theta_{\texttt{T2F}}$ and $\theta_{\texttt{F2I}}$ are in Figure 9): (a) represents the instructions (or "system prompt") used for both fine-tuning and inference; (b) includes the demonstrations used for in-context learning during inference; and (c) shows the input and response structures. During fine-tuning, (c) is filled with cardiologist-adjudicated texts, whereas during inference, (c) retains the format presented above to specify the response format. (d) presents the statistics of our collected patient data, which is further subgrouped by gender, age, and arrhythmia classes – Class I: normal arrhythmias. Class II: clinically significant arrhythmias. Class III: life-threatening arrhythmias. Detailed clinical implications are provided in Appendix C.

## 4.2 INSTRUCTION FINE-TUNING

Utilizing cardiologist-adjudicated data, we apply instruction fine-tuning to instill cardiologist-level professionalism into agents $\theta_{\texttt{M2F}}$, $\theta_{\texttt{T2F}}$, and $\theta_{\texttt{F2I}}$. For $\theta_{\texttt{M2F}}$, we selected Llama-3.1-8B as the base model, LLaVA-v1.5-13B for $\theta_{\texttt{T2F}}$, and another Llama-3.1-8B for $\theta_{\texttt{F2I}}$. Each base model is fine-tuned individually, tailored to its specific task using relevant portions of cardiologist-adjudicated data. For instance, as shown in Figure 3-(a)(c), we fine-tune $\theta_{\texttt{M2F}}$ by utilizing system prompts from (a) and cardiologist-adjudicated texts in the format presented in (c), aligning with the metric-to-findings task handled by $\theta_{\texttt{M2F}}$. Let $\theta_{\texttt{Agent}}$ denote the trainable parameters of any LLM agent, with $X$ and $Y$ representing the instructional input and the corresponding LLM responses for one patient, respectively. Given the cardiologist-adjudicated data $\mathcal{D}$, the tuning process can be formulated as follows:

$$\theta_{\texttt{Agent}}^* = \arg\min_{\theta_{\texttt{Agent}}} \mathbb{E}_{(X,Y)\in\mathcal{D}}\mathcal{L}(\theta_{\texttt{Agent}}(X), Y) \tag{2}$$

The goal of instruction fine-tuning in Eq. 2 is to minimize the mean of the summed loss by $\mathbb{E}(\mathcal{L}(\cdot,\cdot))$ given each pair of $(X,Y)$ within $\mathcal{D}$. Specifically, when $\theta_{\texttt{Agent}}$ is $\theta_{\texttt{M2F}}$, we have $X = (\mathcal{M}, \mathcal{B})$ and $Y = \mathcal{F}$. For $\theta_{\texttt{T2F}}$, $X = (\mathcal{T}, \mathcal{B})$ and $Y = \mathcal{F}$. Lastly, for $\theta_{\texttt{F2I}}$, $X = (\mathcal{F}, \mathcal{G})$ and $Y = \mathcal{I}$.

## 4.3 INFERENCE WITH IN-CONTEXT LEARNING AND FACT-CHECKING

As outlined in Section 3.3, ZODIAC's inference process involves a multi-agent approach using the trained agents $\theta_{\texttt{M2F}}$, $\theta_{\texttt{T2F}}$, and $\theta_{\texttt{F2I}}$. First, $\theta_{\texttt{M2F}}$ processes patient metrics ($\mathcal{M}$) and $\theta_{\texttt{T2F}}$ handles ECG tracings ($\mathcal{T}$), together generating findings ($\mathcal{F}$). These findings are then interpreted by $\theta_{\texttt{F2I}}$ as the diagnostic interpretation ($\mathcal{I}$). Each agent leverages in-context learning to enhance diagnostic accuracy, with fact-checking applied at the final step for backward self-correction.

**In-Context Learning.** For each fine-tuned LLM agent, we implement in-context learning using a set of demonstrations (or "demos") containing cardiologist-adjudicated findings and interpretation. The content of each demo is tailored to the specific LLM agent. For instance, demos for $\theta_{\texttt{M2F}}$ include

cardiologist-adjudicated findings, as shown in Figure 3-(b). To ensure that each demo is relevant to the target patient's case, we categorize the patient data by gender, age group, and arrhythmia class, following the subgrouping presented in Figure 3-(d). We then select three demos that match the patient's gender, age group, and arrhythmia class. During inference, the input prompt is a combination of contents shown in Figure 3-(a)(b)(c).

**Fact-Checking.** Fact-checking occurs after $\theta_{\texttt{F2I}}$ generates the final interpretation ($\mathcal{I}$). ZODIAC applies cardiological guidelines ($\mathcal{G}$) to verify whether the findings ($\mathcal{F}$) correctly lead to the interpretation ($\mathcal{I}$) that aligns with $\mathcal{G}$. Since $\mathcal{F}$ and $\mathcal{I}$ are itemized lists, it is convenient to match each item independently. If discrepancies are identified based on $\mathcal{G}$, ZODIAC automatically prompts the corresponding agent to regenerate specific items in $\mathcal{F}$ or $\mathcal{I}$ using instructions derived from $\mathcal{G}$. Due to space constraints, the example of $\mathcal{G}$ and the fact-checking process are provided in Appendix D.

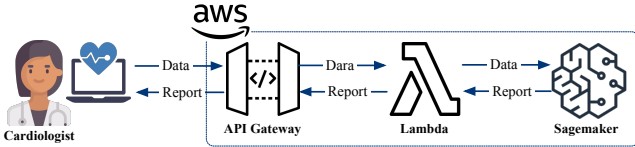

Figure 4: Workflow of ZODIAC assisting cardiologists through AWS deployment.

### 4.4 TOWARDS IN-HOSPITAL DEPLOYMENT

In line with SaMD (FDA, 2018), ZODIAC represents an important milestone in building professional LLMs in today's standard clinical workflow. It has been deployed on Amazon AWS as the backend and connected to the in-hospital frontend to assist cardiologists by providing preliminary reports. As shown in Figure 4, cardiologists or their assistants can upload patient data, including monitored metrics and ECG tracings from wearable patches (Steinhubl et al., 2018) or Holters (Kim et al., 2009). This data is routed through AWS API Gateway, triggering AWS Lambda to invoke SageMaker, where ZODIAC is hosted. On AWS SageMaker, ZODIAC generates preliminary reports, including findings and interpretation based on the data, and performs fact-checking to ensure accuracy before finalizing the reports. These reports are then returned to the cardiologists, who can use them as a foundation to finalize their diagnoses, thus enhancing workflow efficiency and improving diagnostic accuracy.

## 5 EXPERIMENTAL RESULTS

### 5.1 CLINICAL VALIDATION SETTING

Our experiments are designed to align with real-world clinical validation with the following settings:

**Evaluation Metrics:** As detailed in Table 1, we consider eight evaluation metrics commonly used among clinical validations (Tierney et al., 2024; Sallam et al., 2024). Metrics (a)-(e) evaluate the quality of generated outputs from a clinical perspective, while metrics (f)-(h) address potential security concerns. Each metric is rated on a scale from 1 to 5, reflecting varying degrees of alignment with ideal clinical standards.

**Human Validation:** Involving human experts in the validation is critical for enhancing the credibility and acceptance of advanced techniques (Tierney et al., 2024; Sallam et al., 2024). To this end, we engage cardiologists to evaluate ZODIAC using the aforementioned eight metrics. To streamline their assessment process, we developed a structured questionnaire that begins with patient data, followed by generated findings and interpretation, and concludes with rating options (1-5). **Notably, we anonymize the names of the LLMs to ensure fairness, preventing cardiologists from assigning biased scores based on their familiarity with or perceived reputation of specific models, particularly ZODIAC.**

**Dataset:** Instead of using public benchmarks, we adopt real patient data is to align with practical diagnostics. As described in Section 4.1, we use 5% samples from our data collection to validate ZODIAC, reserving the remaining samples for instruction tuning. These samples encompass comprehensive subgroups that align with Figure 3-(d), which we discuss in Section 5.3.

Table 1: Evaluation metrics and their respective domains, abbreviations, and descriptions of ideal cases. Each metric is rated on a scale from 1 to 5, where: 1 — Not at all; 2 — Below acceptable; 3 — Acceptable; 4 — Above acceptable; 5 — Excellent.

| Domain | Evaluation Metrics | Ideal Statements (Findings & Interpretation) are |
|---|---|---|
| Clinics | a) Accuracy (ACC) | statistically correct, aligning with patient's data. |
| | b) Completeness (CPL) | containing complete items to use during diagnostics. |
| | c) Organization (ORG) | well-structured and easier to locate the clinical evidence. |
| | d) Comprehensibility (CPH) | are easier to understand without ambiguity. |
| | e) Succinctness (SCI) | brief, to the point, and without redundancy. |
| Security | f) Consistency (CNS) | mutually supported, without contradicts any other part. |
| | g) Free from Hallucination (FFH) | only containing information verifiable by the guideline. |
| | h) Free from Bias (FFB) | not simply derived from characteristics of the patient. |

Table 2: LLM diagnostic performance across various metrics. Each cell presents "mean (±std)" among ratings from all cardiologists across all patient data. We use **boldface** to indicate the best one.

| Model | Clinic-domain Metric | | | | | Security-domain Metric | | |
|---|---|---|---|---|---|---|---|---|
| | ACC | CPL | ORG | CPH | SCI | CNS | FFH | FFB |
| GPT-4o | 3.6 (1.0) | 4.2 (1.0) | 4.1 (0.7) | 4.2 (0.9) | 4.0 (1.1) | 3.8 (1.1) | 3.9 (1.0) | 4.3 (1.0) |
| Gemini-Pro | 3.7 (1.1) | 4.1 (1.1) | 3.9 (1.0) | 4.0 (1.1) | 4.0 (1.1) | 3.9 (1.1) | 4.3 (1.0) | 4.2 (1.2) |
| Llama-3.1-405B | 3.8 (1.2) | 4.0 (1.0) | 3.9 (1.0) | 4.2 (1.0) | 4.2 (1.2) | 3.8 (1.0) | 4.0 (1.0) | 4.3 (1.0) |
| Mixtral-8x22B | 3.7 (1.1) | 4.1 (1.1) | 4.0 (1.0) | 4.4 (0.9) | 4.2 (0.9) | 4.0 (1.0) | 4.1 (1.0) | 4.4 (0.8) |
| BioGPT-Large | 2.2 (0.4) | 2.8 (0.6) | 3.2 (0.8) | 3.3 (0.7) | 3.2 (0.6) | 3.0 (0.8) | 2.9 (0.7) | 3.8 (0.6) |
| Meditron-70B | 3.3 (1.1) | 3.3 (1.2) | 3.6 (1.1) | 3.6 (1.3) | 3.8 (1.1) | 3.4 (1.2) | 3.3 (1.3) | 3.4 (1.3) |
| Med42-70B | 3.6 (0.9) | 3.8 (0.9) | 3.6 (0.9) | 3.7 (1.1) | 3.7 (1.1) | 4.0 (0.8) | 3.7 (1.0) | 3.6 (1.1) |
| ZODIAC | **4.4 (0.4)** | **4.5 (0.7)** | **4.7 (0.4)** | **4.9 (0.2)** | **4.4 (0.6)** | **4.5 (0.2)** | **4.8 (0.3)** | **5.0 (0.0)** |

**Baselines:** We compare three categories of LLMs: (1) Industry-Leading LLMs: GPT-4o, Gemini-Pro, Llama-3.1-405B, and Mixtral-8x22B. (2) Clinical-Specialist LLMs: BioGPT-Large (Luo et al., 2022), Meditron-70B (Chen et al., 2023b) derived from Llama-2, and Med42-70B (Christophe et al., 2024) derived from Llama-3. (3) Ablations, including a single-agent ZODIAC as detailed in Section 5.4.

**Fair Comparison with Baselines:** For text-based baselines (e.g., Llama-3.1-405B), we employ the same vision-text LLM, LLaVA-v1.5-13B, used in our image-to-text agent. Additionally, the inference-time prompts are identical to those in Figure 3, with one demonstration provided to establish a baseline for basic task understanding.

## 5.2 DIAGNOSTIC PERFORMANCE COMPARISON WITH OTHER LLM PRODUCTS

Comprehensive evaluations in Table 2 highlight the remarkable capabilities of ZODIAC. With fewer than 30B parameters (as noted in Section 4.2), ZODIAC outperforms larger models like Llama-3.1-405B and advanced industrial products such as GPT-4o and Gemini-Pro, particularly in clinical professionalism (e.g., 4.9 CPH) and security assurance (e.g., 5.0 FFB). Additionally, ZODIAC exhibits more stable performance, as evidenced by its lower standard deviation (e.g., ±0.0 FFB). This underscores the importance of incorporating elaborated technical strategies (such as instruction tuning and in-context learning) to enhance diagnostic professionalism, rather than relying solely on prompting a generic model.

Interestingly, medical-specialist LLMs performed worse than generic LLMs. While the small scale of BioGPT-Large (1.5B parameters) understandably limits its diagnostic capabilities, a more critical issue is that the data used for fine-tuning models like Meditron-70B appear to be misaligned with real-world clinical practice. Even when aided by in-context learning demos, these specialist LLMs struggle to meet the specific requirements and security demands of clinical tasks.

Monitoring started on 06/05/2023 04:49:26 and continued for 23:28:22.
- AF/AFL: AF/AFL was not present. No episodes of AFib/Flutter were detected.
- VEB: VEB was present (0.28% burden). The total number of isolated VEB was 286. The longest episode of VEB was not observed.
- VT: VT was not present. No episodes of VT were detected.
- SVEB: SVEB was present (1.15% burden). The total number of isolated SVEB was 1306. The longest episode of SVEB was not observed.
- SVT: SVT was not present. No episodes of SVT were detected.
- Pause: Pause was present. The total number of pauses was 21. The longest pause was 3,596 ms, Day 1 / 20:36:55.
- Block: Block was not present. No blocks were detected.
- Sinus: Sinus rhythm was present (98.5% burden). The predominant rhythm was Sinus Rhythm.
- Symptoms: Symptoms were not present. No symptoms were reported.
- QT interval: The QT interval was 400 milliseconds, which is within the normal range of 350-450 milliseconds.
- PR interval: The PR interval was 100 milliseconds, which is within the normal range of 120-200 milliseconds.
- P-wave: The P-wave was present and had a normal shape and amplitude.
- T-wave: The T-wave was inverted, which is not a normal finding. This may be indicative of a condition called inverted T-wave syndrome.
- ST-segment: The ST segment was within normal limits.

Figure 5: An example of interpretation generated by ZODIAC.

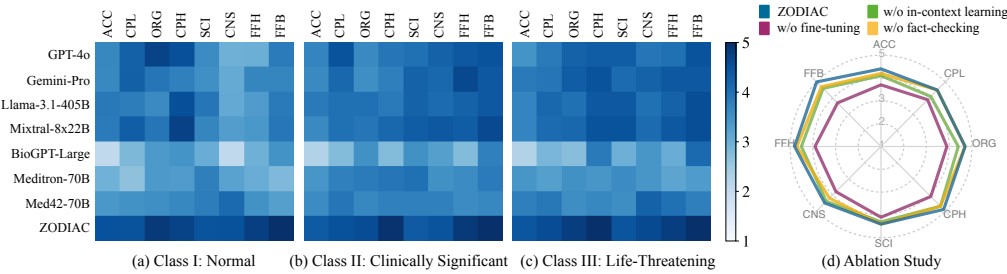

(a) Class I: Normal    (b) Class II: Clinically Significant    (c) Class III: Life-Threatening    (d) Ablation Study

Figure 6: (a)-(c) Subgroup analysis across arrhythmia classes, with the depth of cell color representing rating values (1-5). (d) Ablation baselines.

**Case Study.** Figure 5 presents an example of a ZODIAC-generated interpretation. Compared to other generated results (see Figure 11 to 13), ZODIAC produces accurate and concise statements with clear, structured outputs that are easier for cardiologists to follow. In contrast, other LLMs often generate redundant statements (e.g., GPT-4o in Figure 11 and Gemini-Pro in Figure 12), inaccurate diagnoses (e.g., Llama-3.1-405B in Figure 13), and/or disorganized structures (e.g., GPT-4o and Gemini-Pro), making them more challenging for cardiologists to utilize effectively.

### 5.3 EVALUATING DIAGNOSTIC CONSISTENCY VIA SUBGROUP ANALYSIS

Subgroup analysis is important in clinical validation to evaluate whether a model demonstrates consistent effectiveness across diverse populations (Cook et al., 2004; Rothwell, 2005; Sun et al., 2014). Since our dataset includes various arrhythmia classes, age groups, and genders, we perform evaluations within these subgroups. Figure 6-(a)(b)(c) present results segmented by arrhythmia classes, with additional breakdowns by age and gender shown in Figure 14.

Observe that some industrial products exhibit obviously biased performance across arrhythmias. For example, GPT-4o presents less accuracy (ACC) and completeness (CPL) when diagnosing life-threatening arrhythmias, while Gemini-Pro shows an increase in hallucinations (less FFH) in normal cases, implying imbalances in their pre-trained knowledge. In contrast, ZODIAC delivers consistent performance across all arrhythmia groups, underscoring the importance of data-driven professionalism by incorporating diverse and representative patient cases, as shown in Figure 3-(d).

### 5.4 ABLATION STUDY AND DIAGNOSTIC PROPERTIES OF ZODIAC

**Ablation Study.** We evaluate how removing key components from ZODIAC affects its performance. Figure 6-(d) compares ZODIAC with three baselines: without fine-tuning, without in-context learning, and without fact-checking. Notably, fine-tuning has the greatest impact on diagnostic performance across all metrics, followed by in-context learning, demonstrating the importance of using fine-tuning to permanently embed domain expertise into the LLMs' parameters. Without fine-tuning, in-context learning can still guide the model toward proficiency, but with more limited improvements. We also notice that adding fact-checking improves security-related performance (e.g., FFH, FFB), highlighting the need to integrate clinical guidelines for safe and responsible diagnostics.

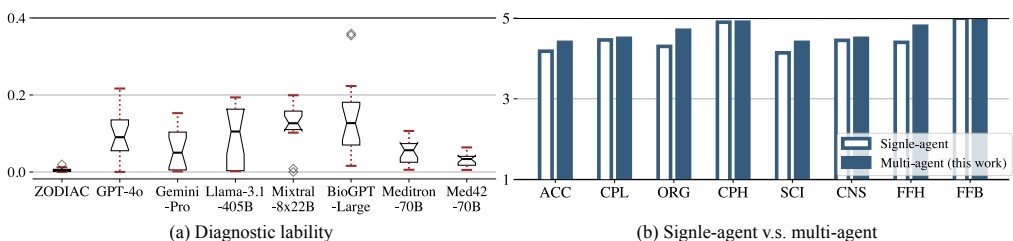

(a) Diagnostic lability                    (b) Signle-agent v.s. multi-agent

Figure 7: Evaluations on (a) diagnostic lability among multiple executions; (b) single-agent ZODIAC.

**Single-Agent Performance.** We evaluate a single-agent variant of ZODIAC. To ensure an equivalent scale, we use the same vision-based LLM, LLaVA, as used in $\theta_{\text{T2F}}$ but with 34B parameters (compared to $\theta_{\text{T2F}}$'s 13B), applying the same tuning and inference techniques to perform tasks originally distributed among multiple agents. However, as shown in Figure 7-(b), we observe clear limitations in performance (e.g., ACC, ORG, SCI, and FFH) for the single-agent ZODIAC. It is important to note that the multi-agent ZODIAC, as described in Section 4.2, comprises fewer parameters in total (30B). This highlights the limited ability of a single LLM to manage various stages of the task, particularly when dealing with different modalities (e.g., table, image, text) and domain-specific expertise. These findings underscore the importance of using collaborative models, especially for complex tasks, as they enhance task-specific proficiency and prevent overloading a single model.

**Diagnostic Lability.** To evaluate the stability of diagnostic outputs, we check if LLMs produce consistent texts (findings and interpretation) across multiple runs. For each patient sample, we run LLMs (ZODIAC or baselines) 10 times. We then convert the generated texts into embeddings using a sentence transformer (Reimers & Gurevych, 2019) and calculate variance of pairwise cosine similarities. The variance serves as the stability score for each patient. Figure 7-(a) shows the stability scores across all patients, and we observe that ZODIAC demonstrates the most stable performance in generating consistent texts over multiple runs. We attribute this stability to the cardiologist-adjudicated texts with consistent structure and content. Through fine-tuning, ZODIAC achieves highly stable outputs with minimal variability across different executions. This level of stability is crucial for ensuring reliability in patient care and building trust among healthcare providers.

# 6 FUTURE WORK

ZODIAC serves as our minimum viable product (MVP) toward functional completeness. Building on ZODIAC, future work focused on **security, trustworthiness, and transparency** is crucial for ensuring long-term success in a competitive market.

As emphasized by FDA's guiding principles (FDA, 2024), securing the development and deployment of LLMs is as important as achieving functional effectiveness. While our current evaluation addresses security-focused metrics, the next phase will prioritize further development of security measures to enhance trust. This will involve investigating third-party adversarial influences in data, identifying inherent weaknesses in LLMs that could lead to vulnerabilities (e.g., backdoors), proposing defensive strategies to safeguard ZODIAC in life-critical diagnostic applications, and promoting transparency to foster human understanding and effective collaboration in human-machine intelligence.

# 7 CONCLUSION

We present ZODIAC, an LLM-powered, multi-agent framework designed for cardiologist-level diagnostics. ZODIAC aims to bridge the gap between clinicians and LLMs in the field of cardiology. Leveraging real-world, cardiologist-adjudicated data and techniques including instruction tuning, in-context learning, and fact-checking, ZODIAC is enhanced to deliver diagnoses with the expertise of human specialists. Through clinical validation, we demonstrate that ZODIAC produces leading performance across patients of different genders, age groups, and arrhythmia classes. In conclusion, ZODIAC represent a significant step toward developing clinically viable LLM-based diagnostic tools.

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

## A    A REAL-WORLD CARDIOLOGICAL REPORT

Figure 8 presents a real-world report on patient data and diagnostics (including findings and interpretation), with identifying information (such as patient name, date of birth, physician name, and company name) anonymized. The report layout is identical to that shown in Figure 2.

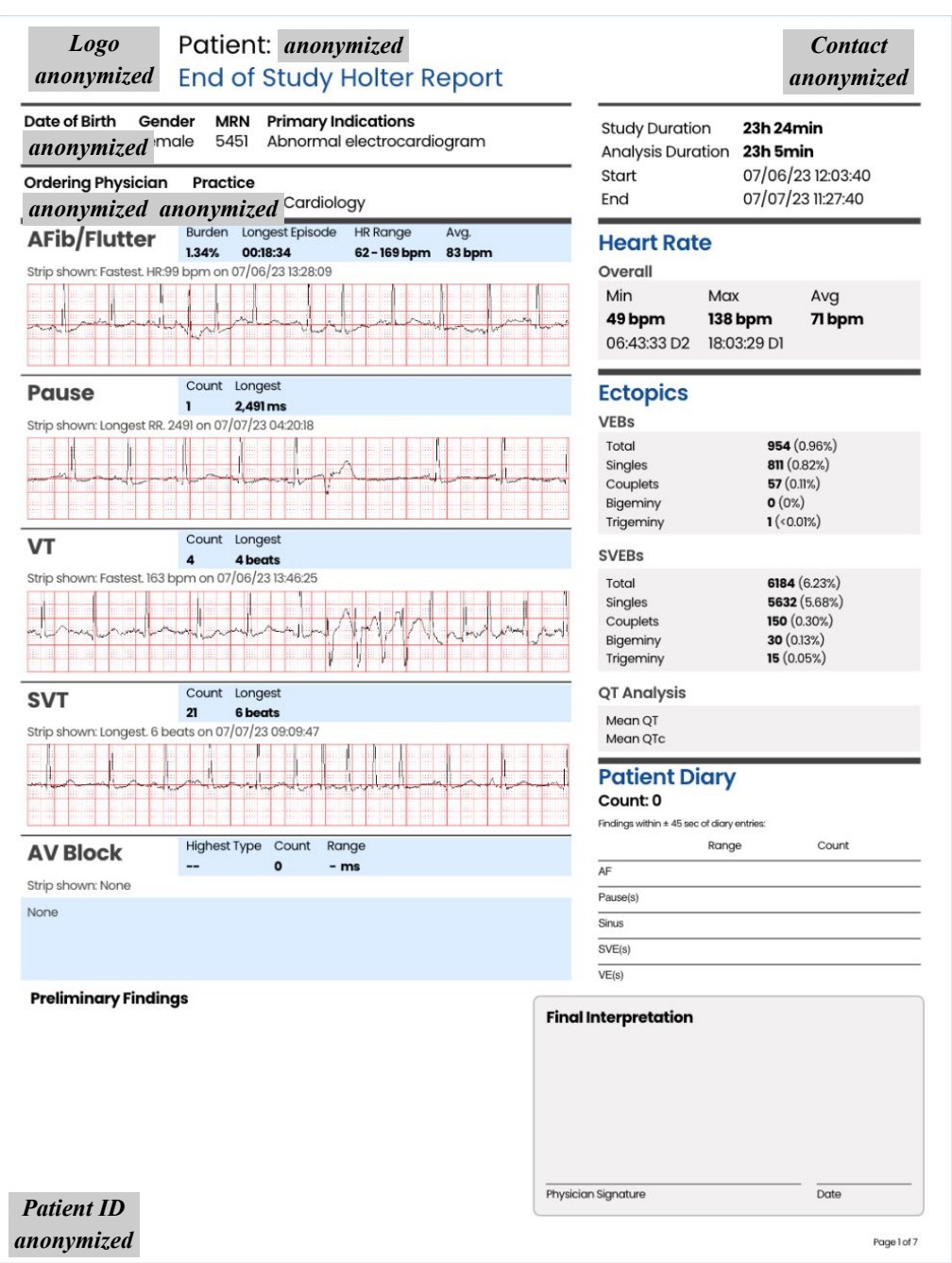

Figure 8: A real-world cardiological report, with identify-related information anonymized.

# B ADDITIONAL PROMPTS

Corresponding to Figure 3-(a)(b)(c), we provide the prompts used for agents $\theta_{\texttt{T2F}}$ and $\theta_{\texttt{F2I}}$ in Figure 9.

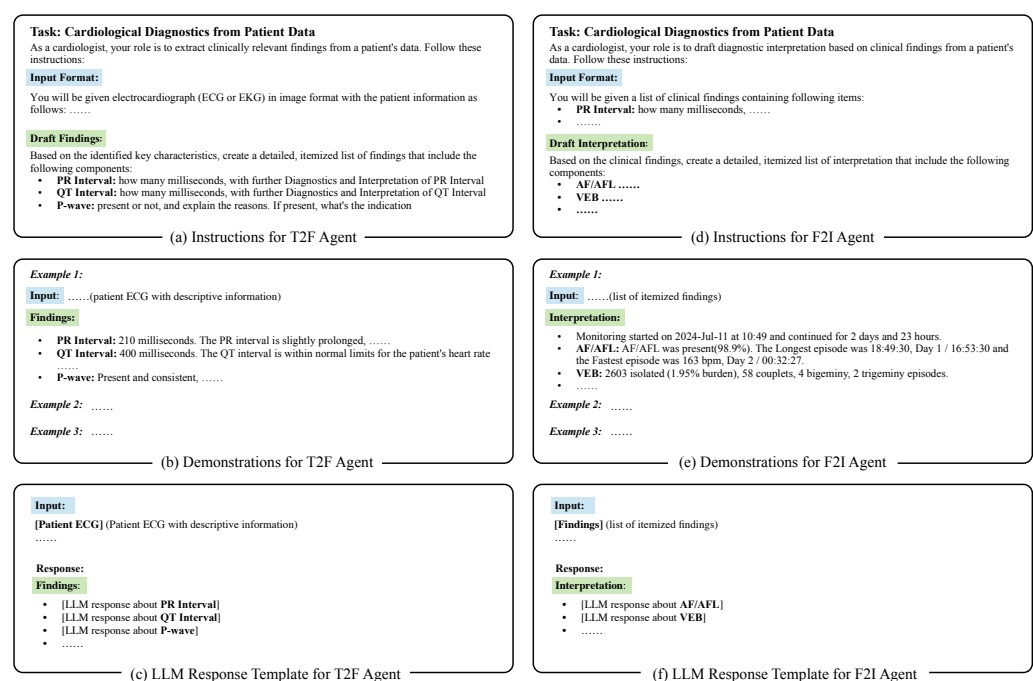

Figure 9: Prompts used for $\theta_{\texttt{T2F}}$ and $\theta_{\texttt{F2I}}$: (a)(d) – instructions or "system prompt"; (b)(e) – demonstrations used during in-context learning; (c)(f) – LLM response template.

# C DETAILS ABOUT ARRHYTHMIA CLASSES

In this work, we categorize arrhythmias into three subgroups:

**Class I — Normal Arrhythmias:** Also known as benign or physiological arrhythmias, these irregular heart rhythms can occur in healthy individuals and typically do not lead to serious health issues. They are generally considered harmless and may not require treatment. In our patient data, Class I arrhythmias include Sinus Bradycardia, Sinus Tachycardia, and Sinus Arrhythmia.

**Class II — Clinically Significant Arrhythmias:** These arrhythmias involve abnormal heart rhythms that can cause symptoms, lead to complications, or require medical intervention. They may disrupt the heart's ability to pump blood effectively, increasing the risk of serious events such as stroke, heart failure, or sudden cardiac death. In our patient data, Class II arrhythmias include Pause (¡3s), Ventricular Premature Beat (PVC), and Atrial Fibrillation (AF).

**Class III — Life-Threatening Arrhythmias:** These abnormal heart rhythms can result in severe consequences, such as cardiac arrest, stroke, or sudden cardiac death, requiring immediate medical attention and often emergency intervention. In our patient data, Class III arrhythmias include Ventricular Flutter (VF), Complete Heart Block (Third-Degree AV Block), Atrial Fibrillation (AFib) with Rapid Ventricular Response, Prolonged Pause, Atrial Flutter (AFL), Ventricular Tachycardia (VT), and Supraventricular Tachycardia (SVT).

In our experiments, we use these arrhythmia classes (I, II, III) for subgroup analysis rather than specific arrhythmias to avoid the limitations of small patient sample sizes for individual conditions. Subgroup analysis based on arrhythmia classes provides a comprehensive view of the LLMs' diagnostic capabilities across different levels of urgency, offering valuable insights for data collection and performance improvement toward more balanced diagnostics.

# D FACT CHECKING USING CLINICAL GUIDELINE

**Clinical guidelines** are systematically developed statements designed to assist healthcare providers and patients in making decisions about appropriate health care for specific clinical circumstances. These guidelines are based on the best available evidence and aim to standardize care, improve the quality of treatment, and ensure patient safety. For example, a section of clinical guidelines about PR Interval is provided in Figure 10.

---

**PR Interval:**

**1. Definition of PR Interval**

    The PR interval measures the period from the onset of atrial depolarization (beginning of the P wave) to the onset of ventricular depolarization (beginning of the QRS complex). It reflects the time taken for the electrical impulse to travel from the sinus node through the atria, AV node, His bundle, bundle branches, and Purkinje fibers to reach the ventricular myocardium.

**2. Range**
- **Normal: 120-200 milliseconds**
- **Prolonged: >200 milliseconds, indicating first-degree AV block**
- **Shortened: <120 milliseconds, may suggest pre-excitation syndromes like Wolff-Parkinson-White syndrome**

**3. Clinical Relevance**
- **Normal PR Interval**
  - ○ **Finding: PR interval within 120-200 milliseconds.**
  - ○ **Interpretation: Indicates normal atrioventricular (AV) conduction. The electrical signal travels from the atria to the ventricles through the AV node and His-Purkinje system within the expected time frame, suggesting healthy cardiac electrical function.**

- **Prolonged PR Interval**
  - ○ **Finding: PR interval longer than 200 milliseconds.**
  - ○ **Interpretation:**
    - ▪ **First-Degree AV Block: The prolongation is uniform across all heartbeats. This is often benign but can be associated with increased vagal tone, intrinsic AV nodal disease, or effects of certain medications (like beta-blockers, calcium channel blockers, or digoxin).**
    - ▪ **Higher degree AV block predisposition: Indicates potential for progression to higher degree AV block, especially in the setting of structural heart disease or acute myocardial infarction.**

- **Short PR Interval**
  - ○ **Finding: PR interval less than 120 milliseconds.**
  - ○ **Interpretation:**
    - ▪ **Pre-excitation Syndromes: Such as Wolff-Parkinson-White (WPW) syndrome where there is an accessory pathway (like the bundle of Kent) allowing premature ventricular activation.**
    - ▪ **Junctional Rhythms: If associated with an abnormal P wave morphology or positioning, may indicate that the impulse originates near or within the AV node rather than the atria.**

- **Variable PR Interval**
  - ○ **Finding: Fluctuating PR intervals across different heartbeats.**
  - ○ **Interpretation:**
    - ▪ **Second-Degree AV Block Type I (Wenckebach): Progressive lengthening of the PR interval until a P wave is not followed by a QRS complex.**
    - ▪ **Atrial Fibrillation with Variable Conduction: If associated with an irregularly irregular rhythm, indicates atrial fibrillation where AV nodal conduction is unpredictably variable.**

- **PR Interval with Grouped Beating**
  - ○ **Finding: Groups of beats with a consistent PR interval followed by a longer pause.**
  - ○ **Interpretation:**
    - ▪ **Second-Degree AV Block Type II: Typically associated with fixed PR intervals on conducted beats, interspersed with non-conducted P waves without prior change in the PR duration.**
    - ▪ **Mobitz Type II or Advanced Block: Often a precursor to complete heart block, requiring immediate assessment and potentially pacing intervention.**

- **Alternating PR Interval**
  - ○ **Finding: Alternation in the length of the PR interval from beat to beat.**
  - ○ **Interpretation:**
    - ▪ **Electrophysiological Variability: May be due to alternating dominance of different AV nodal pathways, a rare phenomenon or related to autonomic tone fluctuations.**
    - ▪ **Underlying Heart Disease: Consider evaluation for ischemic heart disease or infiltrative cardiac conditions that may intermittently affect AV nodal conduction.**

---

Figure 10: Part of clinical guidelines.

**Fact-Checking using Guidelines.** We perform fact-checking by enumerating every itemized finding and corresponding interpretation to identify any misalignment with established guidelines. For example, if the PR interval exceeds 200 milliseconds, the interpretation should include a diagnosis of "a prolonged PR interval, which may indicate a first-degree AV block or the potential for a more advanced block". Failure to include such a diagnosis would signal an inaccurate assessment by ZODIAC. In response, we prompt the relevant LLM agents ($\theta_{T2F}$ and $\theta_{F2I}$ in this case) to re-examine the patient data, verify the accuracy of the findings, and update the interpretation accordingly.

# E    SUPPLEMENTARY EXPERIMENTAL RESULTS

**Case Study.** In line with Figure 5, we present diagnostic outputs from other LLMs using the same patient data, as shown in Figures 11 to 13. Unlike ZODIAC (Figure 5), industrial LLMs tend to produce redundant statements regarding key clinical findings and interpretations, making it difficult for cardiologists to quickly identify key information and make revisions. For example, GPT-4o provides the statement "there were no occurrences of atrial fibrillation or atrial flutter," which could be more succinctly expressed as "AF/AFL: not present"

Moreover, industrial LLMs may present statistical findings without supportive evidence. For instance, Llama-3.1-405B states "VT: not presen" without providing evidence-based interpretation to substantiate the claim, undermining the trustworthiness of the information for cardiologists.

---

**AF/AFL**: There were no occurrences of atrial fibrillation or atrial flutter recorded during the monitoring period (AFib/Flutter Burden: 0%).
**VEB**: The patient demonstrated a total of 318 ventricular ectopic beats (VEB) with a VEB burden of 0.28%. There were 286 isolated VEBs, with the notable presence of 26 VEBs forming ventricular couplets, contributing to a VEB couplet burden of 0.02%. No ventricular bigeminy or trigeminy were recorded.
**VT**: There were no occurrences of ventricular tachycardia (VT Burden: 0%).
**SVEB**: There were 1306 supraventricular ectopic beats (SVEBs) detected, all of which were isolated with a burden of 1.15%. There were no supraventricular couplets, bigeminy, or trigeminy episodes recorded.
**SVT**: No supraventricular tachycardia incidents were recorded during the testing period (SVT Burden: 0%).
**Pause**: The monitoring registered 21 pauses with the longest being 3,596 ms. The total burden of pauses was very low (0.08%).
**Block**: No episodes of atrioventricular (AV) block, including second or third-degree AV block, were observed (Block Burden: 0%).
**Sinus**: The predominant rhythm was sinus rhythm with an average heart rate of 81 bpm. Episodes of sinus tachycardia (552 incidents) and sinus bradycardia (526 incidents) were noted, showing a good balance of sinus dynamics (Sinus Tachycardia Burden: 4.71%, Sinus Bradycardia Burden: 4.23%).
**Symptoms**: No patient-triggered events suggest lack of patient-reported symptoms during the monitoring.
**QT interval**: The recorded QT interval of 400 ms is within the normal range (350-450 ms).
**PR interval**: The recorded PR interval (100 ms) is shorter than the typical range of 120-200 ms, which suggests a potential conduction system anomaly though mild.
**P-wave**: The P-wave is present with normal morphology, indicative of proper atrial depolarization.
**T-wave**: T-wave inversion noted in the report may suggest myocardial ischemia or a repolarization abnormality; further investigation is advisable.
**ST-segment**: The ST-segment appears normal, without elevation or depression indicating no acute ischemic changes during the monitoring time frame. Entering the ECG findings into the clinical context, including background medical history and symptoms, contributes significantly to a comprehensive assessment. Further diagnostic evaluation should be considered to examine the shortened PR interval and the T-wave inversion findings. Regular follow-up and monitoring for any progression of the anomalies mentioned in the ECG findings are recommended.

---

Figure 11: An example of interpretation generated by GPT-4o. The corresponding patient case is same as Figure 5.

---

```
Monitoring started on 06/05/2023 at 04:49:26 and continued for 23:28:22.
- **AF/AFL:** No episodes of Atrial Fibrillation/Flutter were detected during the monitoring period.
- **VEB:** A total of 318 Ventricular Ectopic Beats (VEBs) were detected, with 286 being isolated (burden: 0.25%) and 13
couplet episodes (burden: 0.02%). This indicates a moderate level of ventricular ectopy.
- **VT:** No episodes of Ventricular Tachycardia were detected during the monitoring period.
- **SVEB:** A total of 1306 isolated Supraventricular Ectopic Beats (SVEBs) were detected, with a burden of 1.15%. This
indicates a significant level of supraventricular ectopy.
- **SVT:** No episodes of Supraventricular Tachycardia were detected during the monitoring period.
- **Pause:** A total of 21 pauses were detected, with the longest pause being 3,596 ms on Day 1 at 20:36:55.
- **Block:** No episodes of AV block were detected during the monitoring period.
- **Sinus:** The predominant rhythm was Sinus Rhythm, with a maximum heart rate of 184 bpm, a minimum heart rate of 48 bpm,
and an average heart rate of 81 bpm.
- **Symptoms:** No patient-triggered events were recorded during the monitoring period.
- **QT interval:** The QT interval was 400 milliseconds, which is within the normal range of 350-450 milliseconds.
- **PR interval:** The PR interval was 100 milliseconds, which is shorter than the normal range of 120-200 milliseconds.
This may indicate a potential issue with the electrical conduction system of the heart.
- **P-wave:** The P-wave was present and had a normal shape and amplitude.
- **T-wave:** The T-wave was inverted, which is not a normal finding. This may indicate a potential issue with ventricular
repolarization.
- **ST-segment:** The ST segment was within normal limits.
Overall, the monitoring period revealed a significant level of supraventricular ectopy, moderate ventricular ectopy, and an
inverted T-wave. The PR interval was shorter than normal, which may indicate a potential issue with the electrical
conduction system of the heart. Further evaluation and monitoring may be necessary to determine the underlying cause of
these findings.
```

---

Figure 12: An example of interpretation generated by Gemini-Pro. The corresponding patient case is same as Figure 5.

**Subgroup Analysis.** In alignment with the subgroup analysis presented in Section 5.3, Figure 14 provides additional analysis across different age groups and genders. We observe similar trends

```
Monitoring started on 06/05/2023 at 04:49:26 and continued for 23:28:22.
- AF/AFL: Not present.
- VEB: 286 isolated Ventricular Ectopic Beats were detected (burden: 0.25%). 13 couplet episodes were observed (burden:
0.02%).
- VT: Not present.
- SVEB: 1306 isolated Supraventricular Ectopic Beats were detected (burden: 1.15%).
- SVT: Not present.
- Pause: 21 Pauses were detected, with the longest pause being 3,596 ms.
- Block: Not present.
- Sinus: The predominant rhythm was Sinus Rhythm, with a maximum heart rate of 184 bpm, a minimum heart rate of 48 bpm, and
an average heart rate of 81 bpm.
- Symptoms: No symptoms were reported.
- QT interval: The QT interval was 400 milliseconds, which is within normal limits.
- PR interval: The PR interval was 100 milliseconds, which is slightly shorter than normal.
- P-wave: The P-wave was present and had a normal shape and amplitude.
- T-wave: The T-wave was inverted, which is not a normal finding.
- ST-segment: The ST segment was within normal limits.
Please note that this interpretation is based solely on the provided data and should not be used for diagnostic purposes. A
qualified healthcare professional should be consulted for a comprehensive evaluation.
```

Figure 13: An example of interpretation generated by Llama-3.1-405B. The corresponding patient case is same as Figure 5.

to those described in Section 5.3, where baselines such as GPT-4o, Gemini-Pro, and Llama-3.1-405B exhibit bias by offering better diagnostic completeness (CPL) and comprehensibility (CPH) for adults compared to the elderly population. A similar pattern is observed in the female group, indicating biased pre-training in these LLMs. In contrast, ZODIAC demonstrates more balanced performance across different subgroups, indicating its equitable handling of responsibilities across diverse populations.

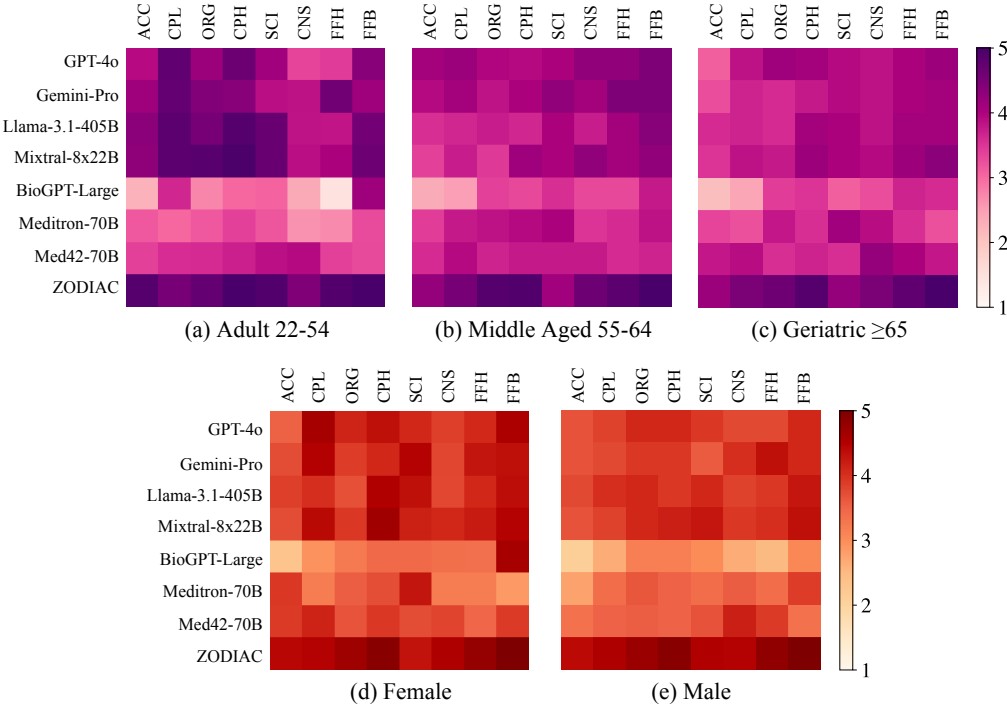

Figure 14: Additional subgroup analysis regarding (a)-(c) age groups and (d)-(e) genders.