# OpenReview forum: "Zodiac: A Cardiologist-Level LLM Framework for Multi-Agent Diagnostics"
_ICLR.cc/2025/Conference — ICLR 2025 Conference Withdrawn Submission_

### Official Review · Reviewer_rBom · 2024-10-30

**Soundness:** 3
**Presentation:** 3
**Contribution:** 3
**Rating:** 6
**Confidence:** 4

**Summary:**

This paper proposes ZODIAC, an LLM-based, multi-agent framework for cardiology diagnostic support. ZODIAC leverages two agents to process clinical metrics (tabular data) and ECG tracings (image data) to generate findings, and a third agent to synthesize the findings into diagnostic interpretations under clinical guidelines. All three agents are instruction fine-tuned on high-quality data collected from expert cardiologists. On eight metrics for clinical validations, ZODIAC outperforms much larger LLMs such as GPT-4o, Gemini-Pro etc. and medical specific LLMs such as BioGPT-Large, Meditron-70B etc., under the judgements of expert cardiologists.

**Strengths:**

1. This paper is clear and well-written.
2. Using LLMs for clinical decision support is a very important topic, which can help reduce clinician burnout and streamline healthcare administration.
3. The design of the ZODIAC framework is intuitive, extensible and flexible, with a correction mechanism with factchecking under clinical guidelines to ensure the safety of the generations.
4. The ablation study is quite comprehensive and justifies the design choice of each component.

**Weaknesses:**

1. In section 4.1, it is mentioned that data from 2000+ patients from collaborating healthcare institutions are used, and 5% of the data is used for evaluation. Given that the evaluation set is in the same distribution with the data used for fine-tuning the LLM agents, I am wondering whether the framework could generalize to a broader patient population, e.g. patients in other hospitals/medical centers, or patients from very different geographic locations or demographics (e.g. different race or ethnicity)?
2. In section 5.1 and table 2, it seems that the numeric results of the eight metrics including accuracy, completeness etc., are aggregated across the ratings from different cardiologists. Could there be any calibration bias across the cardiologists, e.g., some people are much stricter raters and some are more lenient? If that is the case, I think ranking-based metrics are more suitable than directly averaging the ratings across cardiologists.

**Questions:**

1. In line 190, it is mentioned that "$\mathcal{T}$ presents a concise but representative segment of the 24-hour monitoring". Is it possible that there are multiple such segments, e.g. multiple segments with AV block? If so, I think it would be helpful to discuss how the tracings-to-findings (T2F) agent handles multiple ECG images.
2. For equation (2), is the loss $\mathcal{L}$ next-token prediction? I think it would be helpful to clearly mention, or provide the formula, for the loss to help readers better understand.
3. For the evaluation, is it entirely human evaluation? Is there any AI-based evaluation or off-the-shelf metrics? If no AI-based evaluation is used, it may be challenging to scale up the human evaluation and it would be helpful to discuss how to make the human evaluation more scalable or efficient, perhaps in future work.
4. For baseline methods, only one demonstration is provided, while for ZODIAC, three demonstrations which match the patient's demographics and arrhythmia class are used for in-context learning. I am wondering whether this discrepancy in the number and relevance of few-shot examples could contribute to the performance gap between baseline methods and ZODIAC. If that is possible, it may be helpful to provide the baseline methods with the same demonstration examples to isolate this impact.
5. Will the findings and the interpretations data collected from the cardiologists be made public? I think that would be a great contribution to the research community.

---

### Official Review · Reviewer_fR4U · 2024-11-01

**Soundness:** 2
**Presentation:** 3
**Contribution:** 2
**Rating:** 6
**Confidence:** 3

**Summary:**

A well-written work on the creation of an LLM application that can process data from patients, cardiologist reports, and clinical guidelines to perform several tasks.

**Strengths:**

- Well written and nicely presented
- Real-world validation of the approach

**Weaknesses:**

Text:
- typo in figure 7b signle
- line 375: remove "is": Instead of using public benchmarks, we adopt real patient data is to align with practical diagnostics

General:
- *Representative Groups*: race is not indicated as statistic.
- Human validation: how many people have provided a report? Was the data that Zodiac was pre-trained on from the same institution as the physicians? This might explain why clinical-specialist LLM's performed poorly (overfitted to a certain dataset).
- Line 375: *Dataset: Instead of using public benchmarks, we adopt real patient data is to align with practical diagnostics.* Is there a reason you could not do both? Your own benchmark might have some biases as you could have developed your model to perform especially well on the metrics that you defined.
- I am missing a limitations section in your work.
- While I understand that evaluation of these frameworks is far from trivial and physician evaluation is an important part of the solution, more details need to be provided for these experiments. Moreover, public benchmarks should be added. If current benchmarks are imperfect, as mentioned in line 247, a small public benchmark can be provided for future work to be able to compare to your work.
- I urge you to provide code/configurations for as much of your work as possible. This is important, especially to convince the non-health ML field of your work.

**Questions:**

- Line 250: *we utilized ECG data sourced from our collaborating healthcare institutions1 under an IRB-approved protocol, with removed patient identifiers to ensure privacy protection.* The in-practice deployment with AWS makes me think about privacy concerns (even when removing patient indicators, patients can often be easily identified from the remaining information).
- What is the output of the model? From figure 2 it seems that it generates a report. Is this the report from figure 8?

**Details Of Ethics Concerns:**

I think additional details need to be provided on how patient anonymization is guaranteed.

---

### Official Review · Reviewer_LYLg · 2024-11-03

**Soundness:** 1
**Presentation:** 2
**Contribution:** 1
**Rating:** 3
**Confidence:** 4

**Summary:**

The paper introduces ZODIAC, an LLM framework designed to perform diagnostics in cardiology, focusing on multi-agent collaboration for extracting, analyzing, and interpreting patient ECGs. ZODIAC contains three individual agents for transforming metrics to findings, tracing to findings, and findings to interpretation. Through clinical validation on real-world data, the authors claim ZODIAC outperforms leading LLMs. The studied problem is interesting and practical, but there are major issues as listed below.

**Strengths:**

- The studied problem is important and practical in healthcare applications.
- The paper includes human validation and discussion on real-world deployment, which provide insights on how recent advances of LLMs can be utilized in clinical settings.

**Weaknesses:**

- While ZODIAC effectively uses a multi-agent system, the framework is somewhat derivative of existing LLM-based multi-agent approaches. A more comprehensive comparison of this framework against existing multi-agent approaches in healthcare [1,2] is needed to demonstrate the novelty and effectiveness of the proposed framework.

- The experiments are not convinced.
  * The paper relies on a small validation set (only 5% of the dataset, ~100 patients). It raises concerns about the robustness of the framework. The size is far from sufficient to draw solid conclusions about its effectiveness across a broader population or diverse clinical scenarios.
  * All experiments are conducted on single-sourced private data of a limited size (2000+ patients). Though the authors aimed to align real clinical applications by using their own data, there are a bunch of publicly available, widely adopted real-world ECG datasets (e.g., PTB-XL [3], MIMIC-IV-ECG [4], CODE [5]) that can be utilized as additional evaluation sets.
  * There is a lack of out-of-domain evaluation for demonstrating the generalizability of the framework. Both the training data and evaluation data are from the same data source.

- Clinical guidelines are used for fact-checking to generate final interpretations.  However, there is insufficient information on how these guidelines are sourced or managed. There are no details on the selection criteria for clinical guidelines, their integration within the framework, or the mechanisms employed to handle potential conflicts arising from diverse guidelines.

- While the ZODIAC framework leverages a multi-agent setup to mirror cardiological diagnostic processes, the paper does not fully describe how they address potential redundancies or conflicts in findings across different modalities (e.g., ECG and tabular data).

- While the tracing-to-finding agent is intended to interpret ECG images, it remains unclear how crucial this agent is in identifying visual patterns independently, given the availability of comprehensive cardiologist-adjudicated text and clinical guidelines within the framework. This raises the question of whether the model is truly leveraging visual information from ECGs or primarily relying on textual data.

[1] Tang, Xiangru, et al. "Medagents: Large language models as collaborators for zero-shot medical reasoning." arXiv preprint arXiv:2311.10537 (2023).

[2] Bani-Harouni, David, Nassir Navab, and Matthias Keicher. "MAGDA: Multi-agent guideline-driven diagnostic assistance." International Workshop on Foundation Models for General Medical AI. Cham: Springer Nature Switzerland, 2024.

[3] Wagner, Patrick, et al. "PTB-XL, a large publicly available electrocardiography dataset." Scientific data 7.1 (2020): 1-15.

[4] Gow, B., et al. "MIMIC-IV-ECG: diagnostic electrocardiogram matched subset (version 1.0)." PhysioNet (2023).

[5] Ribeiro, Antônio H., et al. "Automatic diagnosis of the 12-lead ECG using a deep neural network." Nature communications 11.1 (2020): 1760.

**Questions:**

See detailed comments above. Briefly,

- How does ZODIAC’s multi-agent framework compare with existing LLM-based multi-agent approaches?

- How robust and generalizable are the conclusions on the framework’s effectiveness across broader and unseen populations?

- How are the clinical guidelines sourced, selected, and integrated into the framework for fact-checking?

- Does the framework primarily rely on textual information, potentially minimizing the tracing2finding agent’s role? In practical scenarios with limited or low-quality cardiologist-adjudicated text, would the framework maintain comparable diagnostic performance relying on ECG tracings?

---

### Official Review · Reviewer_uoEY · 2024-11-03

**Soundness:** 2
**Presentation:** 2
**Contribution:** 2
**Rating:** 3
**Confidence:** 4

**Summary:**

This paper presents ZODIAC, a human-in-the-loop multi-agent framework for electrocardiograms (ECGs) that leverages real-world patient data and cardiologist-adjudicated findings. ZODIAC includes multi-agent collaboration, where the LLM agent extracts findings from either metrics and tracings and extracts interpretation from the combined findings using clinical guidelines. Cardiologists evaluate the generated response with eight different metrics, where ZODIAC shows the best performance with stable outputs evaluated with diagnostic lability. The proposed framework brings commendable clinical rigor to the input data; however, the evaluation lacks sufficient rigor to thoroughly test ZODIAC and its proposed metrics, raising questions about the robustness of its findings. I am inclined to reject the paper in its current form but would be open to revisiting this decision if the concerns and questions are addressed comprehensively during the rebuttal process.

**Strengths:**

- The method and problem are well-defined, and the paper is well-written.
- The proposed approach is built on cardiologist-adjudicated text inputs and is evaluated by clinicians, which enhances its clinical rigor.
- The method incorporates ECG data along with relevant metadata and integrates current clinical guidelines to generate comprehensive reports.

**Weaknesses:**

While the proposed method, ZODIAC, addresses a well-defined clinical problem of generating ECG reports and integrates valuable clinical insights through multi-agent collaboration, concerns remain regarding the rigor of its evaluation.

1. Evaluation Metric: The metrics defined in Table 1 are well-defined for qualitative evaluation, but appear subject to evaluator variability, raising concerns about the objectivity and rigor of its scale in Table 2. The following points are regarding the rigor of this evaluation method which would be expected to be improved in the following submission of the paper.

- Subjectivity of the Metric: The degree of subjectivity within the metrics may not be fully addressed. For example, what are the precise criteria for defining hallucination and bias within the FFH metric? Furthermore, the definition of bias is unclear regarding which 'characteristics' of the patient are considered. Does this refer to patient demographics, clinical features, or another criterion?

- Calibration of the Metric: There are also concerns about whether this metric can serve as a reliable, single quantitative standard for evaluating model outputs. A rigorous calibration process, such as Inter-Rater Reliability (IRR), would help substantiate the metric’s consistency and could address potential variability. This is particularly important because different evaluators may interpret the scale differently—for instance, what a score of 3 represents could vary among physicians.

- Details on Inter-Rater Reliability (IRR): The paper leaves IRR and confidence intervals unexplored, which could suggest inconsistencies in clinical outputs. Addressing IRR would help to ensure that ratings are stable and comparable across evaluators. There are standard deviations reported in Table 2, but it is better to separately report inter-rater confidence intervals as well.

2. Ablation Studies on $\theta_{M2F}$ only or $\theta_{T2F}$ only needed to see the effect of Triple agent vs. Double agents and the effect of leveraging metadata and ECG tracings.

3. There are more recent works on multi-agent collaboration or interaction, which could be included in the related works section of the paper.

[1] Kim, Y., Park, C., Jeong, H., Chan, Y. S., Xu, X., McDuff, D., ... & Park, H. W. (2024). Adaptive Collaboration Strategy for LLMs in Medical Decision Making. arXiv preprint arXiv:2404.15155.

[2] Jin, Q., Wang, Z., Yang, Y., Zhu, Q., Wright, D., Huang, T., … & Lu, Z. (2024). AgentMD: Empowering Language Agents for Risk Prediction with Large-Scale Clinical Tool Learning.

[3] Li, J., Wang, S., Zhang, M., Li, W., Lai, Y., Kang, X., ... & Liu, Y. (2024). Agent hospital: A simulacrum of hospital with evolvable medical agents.

[4] Fan, Z., Tang, J., Chen, W., Wang, S., Wei, Z., Xi, J., ... & Zhou, J. (2024). Ai hospital: Interactive evaluation and collaboration of llms as intern doctors for clinical diagnosis.

[5] Yan, W., Liu, H., Wu, T., Chen, Q., Wang, W., Chai, H., ... & Zhu, L. (2024). ClinicalLab: Aligning Agents for Multi-Departmental Clinical Diagnostics in the Real World.

**Questions:**

Throughout this work, ECG images have been used rather than the raw signal; Is there a specific reason why image instead of signal has been used? I am aware that, in some work, using 2D images has been proven to show higher performance than using 1D signal but curious if that was the case in this work as well.

Wu, Y., Yang, F., Liu, Y., Zha, X., & Yuan, S. (2018). A comparison of 1-D and 2-D deep convolutional neural networks in ECG classification. arXiv preprint arXiv:1810.07088.

---

### Note · Authors · 2024-11-18

I have read and agree with the venue's withdrawal policy on behalf of myself and my co-authors.